# FAP-Specific Signalling Is an Independent Diagnostic Approach in ACC and Not a Surrogate Marker of MRI Sequences

**DOI:** 10.3390/cancers14174253

**Published:** 2022-08-31

**Authors:** Dawn P. Liew, Manuel Röhrich, Lisa Loi, Sebastian Adeberg, Mustafa Syed, Ewgenija Gutjahr, Heinz Peter Schlemmer, Frederik L. Giesel, Martin Bendszus, Uwe Haberkorn, Daniel Paech

**Affiliations:** 1Department of Nuclear Medicine, University Hospital Heidelberg, 69120 Heidelberg, Germany; 2Division of Radiology, German Cancer Research Center (DKFZ), 69120 Heidelberg, Germany; 3Department of Radiation Oncology, University Hospital Heidelberg, 69120 Heidelberg, Germany; 4Department of Pathology, University Hospital Heidelberg, 69120 Heidelberg, Germany; 5Department of Nuclear Medicine, University Hospital Düsseldorf, 40225 Düsseldorf, Germany; 6Department of Neuroradiology, University Hospital Heidelberg, 69120 Heidelberg, Germany; 7Translational Lung Research Center Heidelberg (TLRC), Part of the German Center for Lung Research DZL, 69120 Heidelberg, Germany; 8Clinical Cooperation Unit Nuclear Medicine, German Cancer Research Center (DKFZ), 69120 Heidelberg, Germany; 9Clinic for Neuroradiology, University Hospital Bonn, 53175 Bonn, Germany

**Keywords:** fibroblast activation protein, FAP, adenoid cystic carcinoma, ACC, positron emission tomography, pixelwise correlation, MRI

## Abstract

**Simple Summary:**

Fibroblast Activation Protein (FAP) is a new target for positron emission tomography and computed tomography (PET/CT) imaging of epithelial tumours, such as adenoid cystic carcinomas (ACCs). The current gold standard for ACC imaging is contrast enhanced (ce) MRI, where intertumoural heterogeneity leads to variable appearance on T1-weighted (T1w) and T2-weighted (T2w) images. The correlation of ^68^Gallium (^68^Ga)-labelled FAP-Inhibitor (FAPI) PET signals and MRI signals (contrast-enhanced T1w and T2w) of ACCs has not been described yet. Here, we analysed the correlation of ^68^Ga-FAPI PET signals and MRI signals of 12 ACC patients. ^68^Ga-FAPI PET signals showed a very weak positive correlation with ceT1w values (pooled correlation 0.114, 0.147 and 0.162 at 10, 60 and 180 min) and a weak negative correlation with T2w values. These results underline that ^68^Ga-FAPI PET signalling is not only a surrogate marker of MRI sequences but an independent signal in ACC patients.

**Abstract:**

Background: Fibroblast Activation Protein (FAP) is a new target for positron emission tomography and computed tomography (PET/CT) imaging of epithelial tumours embedded in a fibrous stroma. Adenoid cystic carcinomas (ACCs) have shown elevated tracer uptake in ^68^Gallium (^68^Ga)-labelled FAPIs in previous studies. The current gold standard for ACC imaging is contrast-enhanced (ce) MRI, where intertumoural heterogeneity leads to variable appearance on T1-weighted (T1w) and T2-weighted (T2w) images. In this retrospective analysis, we correlated ^68^Ga-FAPI PET signalling at three time points with ceT1w and T2w MRI signals to further characterise the significance of ^68^Ga-FAPI uptake in ACCs. Methods: Clinical PET/CT scans of 12 ACC patients were performed at 10, 60 and 180 min post i.v. administration of ^68^Ga-labelled-FAPI tracer molecules. ^68^Ga-PET- and corresponding MRI-scans were co-registered, and 3D volumetric segmentations were performed on ceT1w and T2w lesions of co-registered MRI slides. Signal intensity values of ^68^Ga-FAPI PET signalling and ceT1w/T2w MRI scans were analysed for their pixelwise correlation in each patient. Pooled estimates of the correlation coefficients were calculated using the Fisher z-transformation. Results: ^68^Ga-FAPI PET signals showed a very weak positive correlation with ceT1w values (pooled correlation 0.114, 0.147 and 0.162 at 10, 60 and 180 min) and a weak negative correlation with T2w values (pooled correlation −0.148, −0.121 and −0.225 at 10, 60 and 180 min). Individual r-values at 60 min ranged from −0.130 to 0.434 in ceT1w and from −0.466 to 0.637 in T2w MRI scans. Conclusion: There are only slight correlations between the intensity of ^68^Ga-FAPI PET signals and tumour appearance in ceT1w or T2w MRI scans, which underlines that ^68^Ga-FAPI PET signalling is not a surrogate marker of MRI sequences but an independent signal.

## 1. Introduction

Adenoid cystic carcinoma (ACC) is a rare head and neck malignancy, comprising 1% of all head and neck tumours [1]. It is notorious for its slow but invasive growth, high incidence of distant metastases and early perineural spread [2,3].

Histologically, ACCs are composed of basaloid epithelial cells embedded in a fibrous stroma, displaying three morphological subtypes: cribriform (cell nests containing cystlike spaces), tubular (bilayered tubules infiltrating a tumour stroma) and solid (cell aggregates without pseudocysts or lumina) [1,4,5]. These three histological subtypes often appear simultaneously in varying proportions [6,7].

Magnetic resonance imaging (MRI) persists as the mainstay of ACC imaging due to its high soft tissue contrast and high sensitivity of detection of perineural invasion [8]. However, all three histological subtypes of ACC have a varied appearance on MRI with heterogenous appearance in both contrast-enhanced (ce) T1-weighted (T1w) and T2-weighted (T2w) MRI [9].

Cancer-associated fibroblasts (CAFs) in the tumour stroma play a crucial role in the infiltrative nature and tumour progression of ACC [10,11]. A commonly used marker to identify CAF populations is the serine protease fibroblast activation protein (FAP), which is highly expressed in the majority of CAFs [12] and also in activated fibroblasts in sites of fibrosis, wound healing and inflammation [13].

New tracer molecules based on FAP inhibitors (FAPI) have been developed for positron emission tomography and computed tomography (PET/CT) imaging [14,15]. ^68^Gallium (^68^Ga)-FAPI PET/CT as a new and highly promising imaging modality for ACCs has been demonstrated by our group in a previous publication. We showed that the specific and high uptake of ^68^Ga-FAPI in ACCs led to upstaging in several cases and more accurate radiation planning [16].

The purpose of this work is to correlate the uptake of ^68^Ga-labelled FAPI tracers with ceT1w and T2w MRI signalling by pixelwise analysis. We hypothesised that ^68^Ga-FAPIPET signalling is not only a surrogate marker of MRI sequences but an independent signal. A secondary question is how MRI signal intensities correlate with ^68^Ga-FAPI uptake at different acquisition time points (10, 60 and 180 min post injection).

## 2. Materials and Methods

### 2.1. Patient Cohort

Twelve patients with histologically confirmed adenoid cystic carcinoma were included in this retrospective analysis. Informed consent was obtained according to the regulations of the German Pharmaceuticals Act §13 (2b). These patients were referred by their treating oncologist to assist in diagnostic decision making. A total of 7 patients were treatment-naïve, 4 patients had recurrences after surgical resection and 1 patient had a recurrence after definitive radiotherapy. Table 1 gives an overview of the clinical information which was retrieved from the patient charts.

### 2.2. Radiochemistry

Synthesis and labelling of ^68^Ga-FAPI-2 (*n* = 2), ^68^Ga-FAPI-46 (*n* = 7) and ^68^Ga-FAPI-74 (*n* = 3) followed the methods described by Loktev et al. [14,17]. All solvents and non-radioactive reagents were obtained in reagent grade from abcr (Karlsruhe, Germany), Sigma-Aldrich (Munich, Germany), Acros Organics (Geel, Belgium) and VWR (Bruchsal, Germany) and were used without further purification. The chelator DOTA-PNP (2,2′,2′’-(10-(2-4-nitrophenyl)oxy)-2-oxoethyl)-1,4,7,10-tetraazacyclododecane-1,4,7triyl)triacetic acid) was synthesised according to the protocols published by Mier et al. [18].

Radioactive gallium (^68^Ga) was eluted from a ^68^Ge/^68^Ga generator purchased from Themba Labs (Somerset West, South Africa). Labelling was performed according to previously published protocols [14].

### 2.3. Image Acquisition

A Biograph mCT Flow scanner (Siemens Healthineers, Erlangen, Germany) was used for PET imaging. Scans were performed according to previously published protocols [19,20]. After unenhanced low-dose CT (130 keV, 30 mAs, CARE Dose (Siemens); reconstructed with a soft-tissue kernel to a slice thickness of 5 mm), PET was acquired in three-dimensional mode (matrix, 200 × 200) using FlowMotion (Siemens). The emission data were corrected for randoms, scatter and decay. Reconstruction was performed with ordered-subset expectation maximization using 2 iterations and 21 subsets and Gauss filtration to a transaxial resolution of 5 mm in full width at half maximum. Attenuation correction was performed using the unenhanced low-dose CT data. The injected activity for the ^68^Ga-FAPI examinations was 177–285 MBq (details provided in Table 2), and the PET scans were taken at 10, 60 and 180 min after injection. Three patients did not complete the imaging protocol at the three timepoints due to patient status and/or imaging constraints.

The acquisition of MRI scans was performed at 3T, employing the following previously published parameters [21]: T1-weighted GdCE MRI (GdCE-T1) (echo time (TE) = 4.04 ms; repetition (RM1) time (TR) = 1710 ms; field of view (FoV) in mm2: 256 × 256; matrix: 512 × 512; slice thickness: 1 mm, transaxial resolution 0.85 mm) and T2-weighted MRI turbo spin echo (TSE) (TE = 86 ms; TR = 5550 ms; FoV: 229 × 172 mm^2^; matrix: 384 × 230; slice thickness: 5 mm, transaxial resolution 0.75 mm). The time interval between ^68^Ga-FAPI PET/CT and MRI scans of the head and neck was on average 20.8 days.

### 2.4. Co-Registration of ^68^Ga-FAPI PET/CT and MRI Scans

^68^Ga-FAPI PET/CT scans at all timepoints were co-registered on both T1w and T2w MRI scans, respectively, using a rigid co-registration tool in PMOD (Zurich, Switzerland) and corrected manually if necessary. Regions of interest were manually drawn on transaxial MRI images based on morphological appearance and automatically converted to a 3D volume of interest (VOI). The VOI was corroborated by a board-certified nuclear medicine physician (MR) and board-certified radiologist (DP). The pixel data of ^68^Ga-FAPI PET and MRI signal intensity was extracted using the Pixeldump tool of the PMOD software.

### 2.5. Statistics

The pixelwise analysis of co-registered PET/CT and MRI scans was performed with Fisher *z*-test. Pooled estimates of the Pearson correlation coefficients (r) of all patients at each respective timepoint were calculated using the Fisher z-transformation [22]. We interpreted the Pearson correlation coefficient as suggested by Cohen [23], where r ≈ 0 means no correlation, r ≈ ±0.1 a weak correlation, r ≈ ±0.3 a moderate correlation and r ≈ ±0.5 a strong correlation.

A *p*-value of <0.05 was defined as statistically significant. All statistical analyses were performed using Microsoft Excel 2010.

## 3. Results

### 3.1. Description of the Results

#### 3.1.1. Very Weak Positive Correlation of ^68^Ga-FAPI PET and CeT1w Signalling in ACCs

^68^Ga-FAPI PET uptake showed a very weak positive correlation with ceT1w signal intensity (pooled correlation 0.114, 0.147 and 0.162 at 10, 60 and 180 min). Pooled r-values at 10 min (−0.312 to 0.470), 60 min (−0.130 to 0.434) and 180 min (−0.112 to 0.452) are displayed in Table 3. The correlation of ^68^Ga-FAPI uptake and ceT1w signal intensity had a slight tendency towards increasing positive correlation coefficients (Figure 1). Previous treatment does not appear to affect the correlation of ^68^Ga-FAPI uptake and ceT1w signal intensities at 60 min (treated: z corr r = −0.080–0.354; treatment-naïve: z corr r = −0.130–0.434).

#### 3.1.2. Weak Negative Correlation of ^68^Ga-FAPI PET and T2w Signalling in ACCs

^68^Ga-FAPI PET uptake showed a weak negative correlation with T2w signal intensity (pooled correlation −0.148, −0.121 and −0.225 at 10, 60 and 180 min). Pooled r-values at 10 min (−0.559 to 0.156), 60 min (−0.466 to 0.637) and 180 min (−0.462 to −0.071) are displayed in Table 4.

Correlation of ^68^Ga-FAPI uptake and T2w signal intensity does not seem to have any discernible trend across time (Figure 1). Treatment does not appear to affect the correlation of ^68^Ga-FAPI uptake and T2w signal intensities (treated: z corr r = −0.466–0.637; treatment-naïve: z corr r = −0.453–0.125).

Figure 2 shows an example of a primary ACC manifestation (Patient 4) in the right nasal sinus. In ceT1w and T2w MRI sequences, the tumour appears largely homogenous. However, ^68^Ga-FAPI uptake is surprisingly incongruent, with hotspots in areas that have no discernible difference in appearance in MRI. This is reflected in the low correlation value (ceT1w at 60 min: 0.173; T2w at 60 min: −0.264).

Figure 3 comprises exemplary correlation scatter plots (Patient 2) of ceT1w and T2w at all three timepoints (10, 60 and 180 min). In this case, there is a weak positive correlation between ceT1w and ^68^Ga-FAPI uptake and no correlation between T2w and ^68^Ga-FAPI uptake.

## 4. Discussion

In this retrospective voxelwise analysis of ^68^Ga-FAPI PET uptake and MRI signal intensities of 12 ACCs, we found slight correlations between ^68^Ga-FAPI PET signals and tumour appearance in ceT1w or T2w MRI scans. This indicates that ^68^Ga-FAPI PET is an independent imaging method for ACCs and not only a surrogate signal of the clinically established MRI sequences. This result is in line with our previously published analyses of ^68^Ga-FAPI PET and MRI signalling in glioblastomas, likewise correlating ^68^Ga-FAPI uptake with MRI sequences (relative cranial blood volume (rCBV) and apparent diffusion coefficient (ADC) using similar methodology, where we concluded that ^68^Ga-FAPI PET signalling does not only reflect intratumoural perfusion differences nor cell density [21]. Similarly, we had observed differential spatial distribution of intensely ^68^Ga-FAPI-positive spots and the contrast enhancing lesion of glioblastomas [24]. FAPI-positive spots in glioblastomas may reflect tumour areas with increased invasiveness or epithelial to mesenchymal transition, as it has been demonstrated previously that overexpression of FAP contributes to these processes [25].

The correlation coefficients of ^68^Ga-FAPI PET and MRI remained relatively stable when different acquisition time points for ^68^Ga-FAPI PET were compared. Several previous studies have analysed different ^68^Ga-FAPI PET acquisition time points ranging from 10 min to 3 h for various tumour entities, including ACC, and found decreasing signal intensities but increasing tumour-to-background ratios (TBR) over time, so that the optimal imaging time point for ^68^Ga-FAPI PET is still a point of discussion [16,26,27,28,29]. In our dataset, ^68^Ga-FAPI PET signal intensities decreased over time, too, while the correlations of 68Ga-FAPI PET images with MRI sequences were mostly stable when comparing PET acquisition at 10, 60 and 180 min p.i. This underlines that different acquisition time points of FAPI PET result in comparable imaging information.

With respect to the imaging and clinical management of ACC, ^68^Ga-FAPI PET has been shown to deliver high-contrast images of ACC tumours and can sensitively detect perineural invasion as well as skip lesions and metastatic lesions, leading to changes in TNM-based staging and radiation plans in a significant portion of ACC patients [16]. Similar results have been reported for other head and neck cancers [30,31]. On the other hand, MRI represents the gold standard for clinical imaging of ACCs, whereas ceT1w and T2w sequences are crucial for the detection of ACC manifestations [32,33]. However, MRI-based morphological information is not fully conclusive in all cases as other conditions such as infection, inflammation, trauma, vascular lesions and haematoma as well as constitutive contrast enhancement of cranial nerves may mimic the MRI appearance of ACCs [34]. Especially in post-operative and post-radiotherapy settings, differentiation between post-therapeutical changes and tumour relapse can be challenging [35].

Promising results have been reported for hybrid PET–MRI using ^18^Fluor-Fluorodeoxyglucose (^18^F-FDG), which improved the diagnostic accuracy of recurrent ACCs compared with MRI alone [35]. Similarly, ^68^Ga-FAPI PET could be used for future hybrid imaging MRI/PET approaches of ACCs, especially as ^68^Ga-FAPI PET has been shown to outperform ^18^F-FDG in many malignancies, especially in head and neck tumours due significantly to lower background signalling in the skull base and head and neck region [36,37].

Our findings underline the independent imaging information provided by ^68^Ga-FAPI PET compared with MRI sequences for ACCs. However, considering the heterogeneity of the examined patient population (treatment-naïve and recurrent ACCs), as well as differences in histological subtypes, we recommend investigations of ^68^Ga-FAPI PET and MRI with larger and stratified ACC cohorts in order to further evaluate the high potential of both imaging methods for ACCs. Future studies should investigate the diagnostic accuracy for the differentiation of ACC and other tumours of the head and neck region, the influence of therapy-associated changes in the appearance of ACCs in MRI and FAPI PET and the prognostic value of the imaging modalities. This was not possible in our investigation due to its retrospective setting and the limited number of patients. A minor limitation of our analysis is the fact that different FAPI tracer variants were used for PET imaging. We have shown for several cancer entities that these tracer variants do not significantly differ in their imaging properties [17,26,38]. Beyond that, the pharmacokinetics of FAPI-tracers may have had a certain influence on the correlations with MRI signalling which we have found. However, it has been described that FAPI-02, FAPI-46 and FAPI-74 all show relatively stable uptake over time between 10 min and 180 min in various tumours, including ACCs [16,26,39], so that we consider the potential influence of the tracer kinetics on our results as small. Another minor limitation of the study is that the used ceT1w and T2w imaging data cannot not be regarded as quantitative metrics, however, relative signal associations can still be assessed with our approach.

## 5. Conclusions

Standardised uptake values in ^68^Ga-FAPIPET correlate only slightly with ceT1w and T2w MRI signal intensities in ACCs, which indicates that ^68^Ga-FAPI PET is not a surrogate signal of the clinically established MRI sequences. ^68^Ga-FAPI PET/CT is a independent imaging modality providing complementary information to conventional MRI for ACCs.

## Figures and Tables

**Figure 1 cancers-14-04253-f001:**
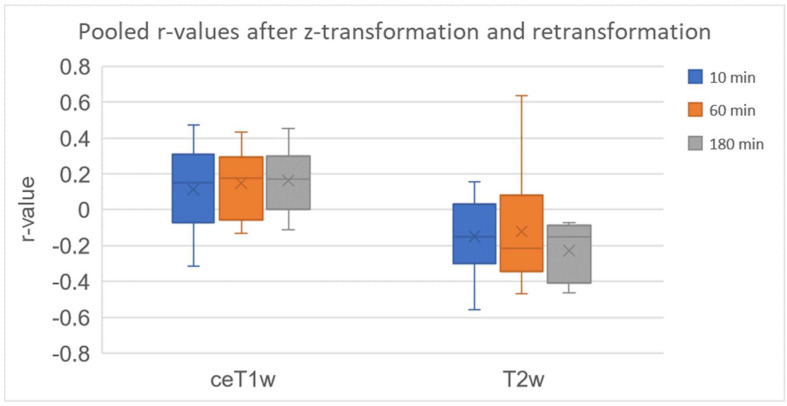
Pooled r-values of the correlation between ^68^Ga-FAPI PET and MRI sequences (contrast enhanced T1w and T2w) after z-transformation and retransformation at all three timepoints. Range of pooled r-values (ceT1w): 10 min (−0.312 to 0.470), 60 min (−0.130 to 0.434) and 180 min (−0.112 to 0.452). Range of pooled r-values (T2w): 10 min (−0.559 to 0.156), 60 min (−0.466 to 0.637) and 180 min (−0.462 to −0.071).

**Figure 2 cancers-14-04253-f002:**
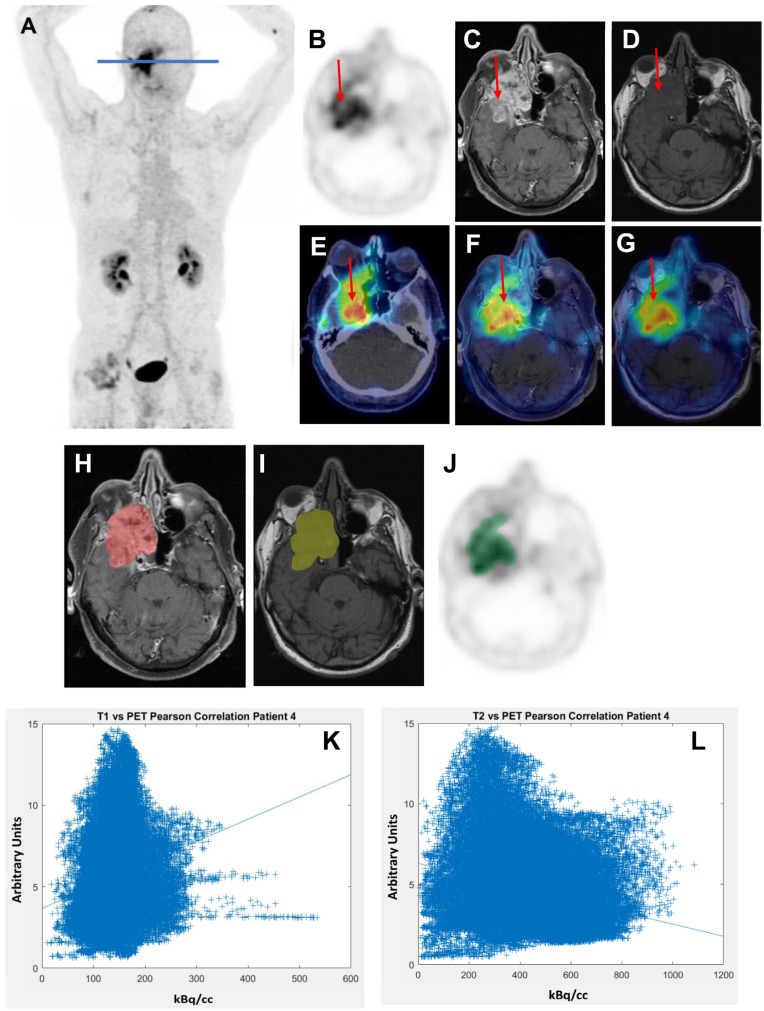
Example (Patient number 4) of a treatment-naïve ACC (tubular type) in the right nasal sinus. (**A**): Maximum Intensity Projection (MIP); (**B**–**G**): appearance of the tumour in different imaging modalities: (**B**): ^68^Ga-FAPI PET, (**C**): MRI (ceT1w); (**D**): MRI (T2w); (**E**): ^68^Ga-FAPI PET fused with CT; (**F**): ^68^Ga-FAPI PET fused with ceT1w MRI, (**G**): ^68^Ga-FAPI PET fused with T2w MRI. Red arrows in B–G point at the tumour. (**H**–**J**): Manual tumour delineation based on (**H**): ceT1w MRI (tumour delineation in red), (**I**): T2w MRI (tumour delineation in yellow) and (**J**): ^68^Ga-FAPI PET (tumour delineation in green). (**K**): ceT1w correlation scatter plot at 60 min; (**L**): T2w correlation scatter plot at 60 min.

**Figure 3 cancers-14-04253-f003:**
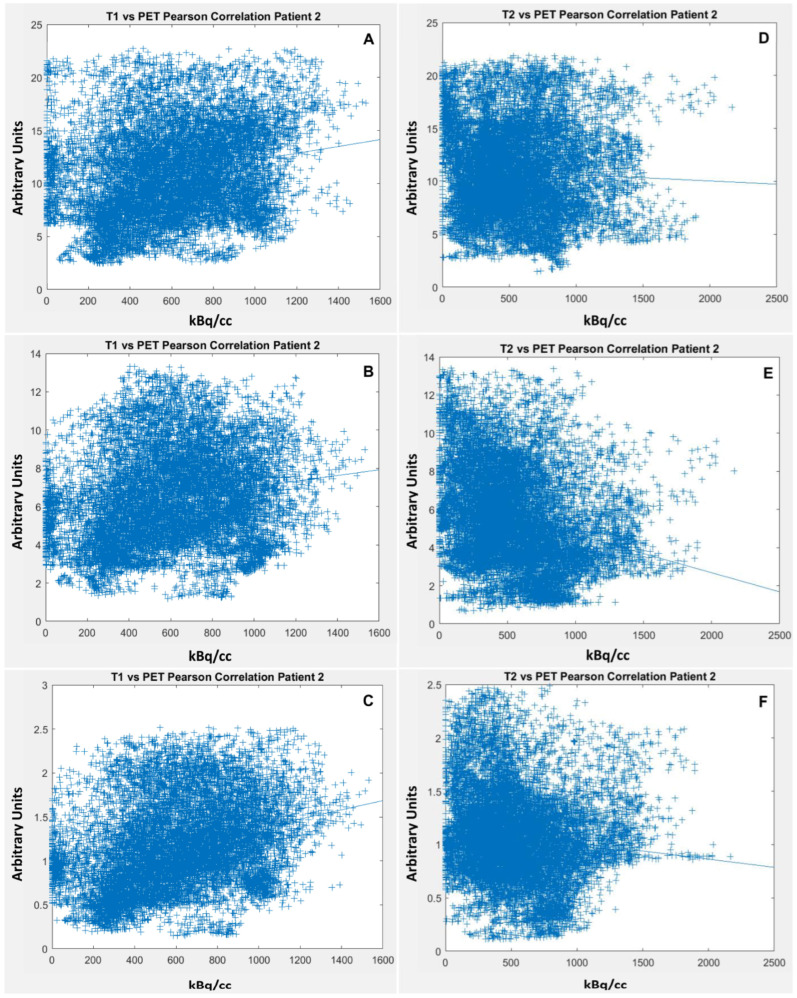
Patient number 2: Left column (**A**–**C**): correlation of ceT1w vs. ^68^Ga-FAPI PET at 10, 60 and 180 min. Right column (**D**–**F**): correlation of T2w vs. PET/CT at 10, 60 and 180 min.

**Table 1 cancers-14-04253-t001:** Clinical and histological characteristics of 12 patients with ACC.

Patient Number	Age (Years)	Sex (M/F)	Location of Primary Tumour	Clinical Setting (and Duration of Disease)	Previous Therapy	Histological Subtype
1	29	F	Right parapharyngeal space	Primary	None	Mixed (tubular and cribriform)
2	71	F	Left maxillary sinus	Recurrence (5 months)	Resection	Cribriform
3	56	M	Left oropharynx	Recurrence (6 years)	Resection, neck dissection, re-resection	Not available
4	71	M	Right maxillary sinus	Primary	None	Tubular
5	34	F	Right nasopharynx	Primary	None	Not available
6	66	F	Left maxillary sinus	Recurrence (4 years)	Resection	Cribriform
7	53	F	Right hard palate	Recurrence (5 years)	Definitive radiotherapy	Mixed (tubular and cribriform)
8	69	F	Left parotid gland	Recurrence (28 years)	Resection, re-resections, resections of pulmonary metastases, Nivolumab	Cribriform
9	63	M	Floor of the mouth	Primary	None	Cribriform
10	66	F	Epi- and oropharynx, median	Primary	None	Tubular
11	48	M	Right parapharyngeal space	Primary	None	Cribriform
12	68	F	Epipharynx, left base of the skull	Primary	None	Cribriform

**Table 2 cancers-14-04253-t002:** ^68^Ga-FAPI PET/CT imaging protocols of 12 patients with ACC.

Patient Number	FAPI Tracer	Injected Activity (MBq)	SUV_max_			SUV_mean_		
			10 Min	60 Min	180 Min	10 Min	60 Min	180 Min
1	FAPI-2	258	-	16.8	-	-	7.73	-
2	FAPI-74	269	6.15	6.24	3.84	3.81	2.58	2.32
3	FAPI-74	221	10.2	8.15	4.27	4.63	3.37	2.75
4	FAPI-74	177	12.0	11.2	6.71	7.31	4.07	3.97
5	FAPI-46	282	14.0	12.8	7.95	6.99	4.65	4.36
6	FAPI-46	285	-	17.6	-	-	8.05	-
7	FAPI-46	220	9.17	7.94	7.85	4.63	4.55	3.93
8	FAPI-2	268	-	4.26	-	-	2.92	-
9	FAPI-46	211	9.64	10.7	13.3	4.59	5.53	4.92
10	FAPI-46	269	12.2	13.1	12.2	8.08	6.52	5.53
11	FAPI-46	268	10.8	10.7	13.4	6.63	5.35	4.08
12	FAPI-46	244	12.1	14.4	-	4.99	5.83	-

**Table 3 cancers-14-04253-t003:** Correlation results of 68Ga-FAPI uptake and ceT1w MRI signal.

Patient Number	Number of Pixels	Correlation at 10 Min	*p*-Value	Correlation at 60 Min	*p*-Value	Correlation at 180 Min	*p*-Value
1	7892	-	-	0.172	<0.001	-	-
2	12,118	0.226	<0.001	0.181	<0.001	0.319	<0.001
3	77,987	0.470	0	0.312	0	0.093	<0.001
4	124,187	0.150	0	0.173	0	0.175	0
5	37,870	0.038	<0.001	0.246	0	0.243	0
6	177,586	-	-	−0.080	<0.001	-	-
7	151,081	−0.156	0	−0.130	0	−0.112	0
8	2531	-	-	0.354	<0.001	-	-
9	19,175	−0.312	0	−0.076	<0.001	−0.029	<0.001
10	30,090	0.210	<0.001	0.195	<0.001	0.165	<0.001
11	18,669	0.394	0	0.434	0	0.452	0
12	9654	0.010	0.351	−0.006	0.533	-	-

**Table 4 cancers-14-04253-t004:** Correlation results of 68Ga-FAPI uptake and T2w MRI signal.

Patient Number	Number of Pixels	Correlation at 10 Min	*p*-Value	Correlation at 60 Min	*p*-Value	Correlation at 180 Min	*p*-Value
1	11,903	-	-	0.124	<0.001	-	-
2	14,005	−0.048	<0.001	−0.263	<0.001	−0.122	<0.001
3	111,994	0.155	0	0.090	<0.001	−0.071	<0.001
4	81,879	−0.175	0	−0.264	0	−0.432	0
5	8098	−0.401	<0.001	−0.424	0	−0.392	0
6	47,534	-	-	−0.351	0	-	-
7	160,768	−0.072	<0.001	−0.110	0	−0.133	0
8	458	-	-	0.563	<0.001	-	-
9	18,432	−0.508	0	−0.435	0	−0.364	0
10	13,520	0.112	<0.001	0.051	<0.001	−0.074	<0.001
11	12,597	−0.150	<0.001	−0.230	<0.001	−0.166	<0.001
12	12,586	−0.172	<0.001	−0.193	<0.001	-	0

## Data Availability

The data presented in this study are available in this article.

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
