# Peer review of "FAP-Specific Signalling Is an Independent Diagnostic Approach in ACC and Not a Surrogate Marker of MRI Sequences"

_cancers, 2022, doi:10.3390/cancers14174253_

Round 1
Reviewer 1 Report
In the submitted manuscript "FAP-specific signalling is an independent diagnostic approach 2 in ACC and not a surrogate marker of MRI sequences", the authors have briefly introduced the co-relation studies of a newly introduced imaging technique called 68Ga-FAPI PET signaling and the routinely practiced techniques of MRI signals in cancer patients, especially focusing on adenoid cystic carcinomas (ACCs). The analysis was conducted and concluded only slight correlations exist between 68Ga-FAPI PET signals MRI signals
Minor suggestions:
1) It is suggested to spellcheck the manuscript and set it to either American-English accepted word formats or other formats. For example, the word ‘signalling’ or is it ‘signaling’; ‘tumour’ or is it ‘tumor’.
2) In the Results section, it is suggested to highlight the title of the result section, either by an underline or formatting (Line# 144 or Line# 154).
3) In Figure-2 B – E, it is suggested that the author can direct which region the reader should focus on. It can be pointed by an arrowhead or a star.
4) In Figure 2 and Figure 3 – for co-relation plots, it is suggested that the author can provide X-axis and Y-axis titles, which will provide some more information for the readers.
5) In the simple summary section, the author especially mentions adenoid cystic carcinomas but ends with a general statement. It is suggested that the author that “68Ga-FAPI PET signaling is not only a surrogate marker of MRI sequences but an independent signal in ACC patients”
Author Response
Thank you for the good review.

Reviewer 2 Report
Dawn Liew et al. present a new diagnostic approach for patients suffering from ACC based on 68Ga-FAPI PET. This approach have a great potential to provide important additional informations compared to the the contrast enhanced MRi, which is the actual gold standard . Therefore, the authors correlated both approaches and found hot spots in the tumours with increased uptake of 68Ga-FAPI, which were not or not sufficiently detectable by MRi. Hence, correlation between MRi and PET was low or not given.
The authors provide a clear introduction and in principal sound discussion. In contrast, the result part is confusing and not clear presented, with contradicting descriptions.
Minor:
1) Please, make decimal digits uniform (3 should be enough) in text and tables
2) Please, check line 30 "... is a target for Positron...." if there is a space to much; line 43 68Ga -> 68Ga; line 96 ABCR -> abcr?
3) Please, add the information of pixel size/resoltution of MRi images and PET images in methods
Major:
1) It seems there are some half sentences left over? lines 144, 154
2) Give the ranges of pooled r-values at all time points with number in brackets
3) line 146: the correlation is increasing over time, but in line 148: no trend over time? this is not fitting, please explain
4) line 152: patients 4 and 10 show a trend towards stronger positive correlation -> table 3 shows higher correlations for other patients
5) line 154 ceT2w?
6) line 167 to 168 statement is not fitting, compared what is written before or misunderstandable
7) Fig.2 B to E, Please try to use the same plane and orientation in all images and add an overlay/corregistration of ceT1w and PET, as well as T2w and PET and if possible an outlining of the MRi and PET signal extension in the target regions
8) Please, add a statement if PET signal (SUVmax and SUVmean) is affected by treatment and add both SUV values for all patients in Table 1
9) Please, add a statement about the performance/pharmacokinetics of the used 68Ga-FAPI radiotracers and if they could have an impact on the correlations you performed
10) line 210 to 212, there is not really understandable what is meant. MRi sequences (?) and stable correlation with PET is in contradiction with the general idea of an independent marker, Please, be more clear.
Thank you very much for your work and presentation of your interesting findings! Looking forward to your responses.
Author Response
Thank you for your review. Please see the file attached.

Reviewer 3 Report
General problem
Clinical and pathological information is very poor in this small and very heterogeneous cohort of patients.
List of authors
It is not acceptable that there is no pathologist among the authors. This prvents readers from judging who was in charge of the presented diagnoses (considering that ACC has quite some difficult differential diagnoses).
Simple summary
Line 23 The abbreviation FAPI should be introduced at the first use and not later (here line 33)
Abstract
No convincing explained is given, why the poor correlation between two variables indicates that one variable is a surrogate marker of the other AND can indicate that the variables are independent (lines 49-50). In fact the authors contradict each other, when taking their statement in the Conclusion chapter (lines 236-237) into account: there, the authors state that 68Ga-FAPI-PET is NOT a surrogate signal of the clinically established MRI sequences.
Results
Lines 151-153 The authors highlight a trend between 68Ga-FAPI uptake and ceT1w in patients #4 and #10 with tubular ACC. However, in the absence of detailed information beyond the histological ACC subtype it is not reasonable to highlight this fact due to the low number of cases.
Discussion
Lines 217-219: „On the other hand, MRI represents the gold standard for clinical imaging of ACC, whereas the ceT1w and T2w sequences are crucial for the detection of ACC manifestations“. What does it mean when the gold standard is not fulfilling crucial requirements?
Line 226: „Our findings underline the independent diagnostic value…“
Reviewer: This is not correct. The authors show different imaging outcomes but do not show any „diagnostic value“.
The authors do not discuss, whether the images shown are of any help to differntiate ACC from the myriad of other tumors that occur in the indicated anatomic region. It is also not discussed whether different images reflect any tissue-based prognostik factors.
Table 1
ACC can show a cribriform, tubular and solid growth pattern with highly variable proliferation indices. Therefore, it is not clear, what the histological subtype „mixed“ indicates: Is it cribriform-plus-tubular or tubular-plus-solid? Pathological information is extremely poor.
Conclusion
The authors state that „Ga-FAPI-PET/CT is a highly promising independent imaging modality…“. However, no data are provided to show why this is the case, i.e. why it is „promising“. Obviously, the images look different but any clinical relevance of this is not shown.
The discrepancy between the Conclusion and Abstract has been mentioned above.
Author Response
Thank you for your review. Please see the point to point response in the attachment.

Round 2
Reviewer 2 Report
Dear authors,
Thank you very much for adapting your manuscript and addressing all my questions and comments.
Minor:
1) Please, add a LUT for PET and MRi in Fig.2
2) Please, add a short statement about the 3 weeks between PET/CT and MRi in discussion part, if the delay in time could have an influence on the comparability of these data sets.
Thank you very much for your efforts in advance!
Reviewer 3 Report
Thank you for the revision that complies with this revieer's requests. I suggest to correct the wording/spelling of 2 sentences
Line 225-226: Please change to: In our dataset, 68Ga-FAPI -PET signal intensities decreased over time as well, while the correlations…
Line 273: Please change to: …the used ceT1w and T2w imaging data cannot be regarded as quantitative metrics…